# OpenReview forum: "SEC-bench: Automated Benchmarking of LLM Agents on Real-World Software Security Tasks"
_NeurIPS.cc/2025/Conference — NeurIPS 2025 poster_

### Official Review · Reviewer_myBf · 2025-06-22

**Clarity:** 4
**Significance:** 3
**Originality:** 2
**Rating:** 5
**Confidence:** 3

**Summary:**

This paper proposes (a) a method to create
a benchmark for vulnerability detection and
exploit generation and vulnerability repair,
and (b) a specific benchmark constructed in
this way.  It also uses the resulting benchmark
to (c) measure the effectiveness of existing
LLMs and agents at these tasks.  We learn
that current systems are pretty weak at these
tasks.

**Questions:**

Can you help clarify the categories in Section 3.3?
Are there any insights we can learn about the biggest
opportunities for improvement, that might be useful
to improve these agents?

**Ethical Concerns:**

["NO or VERY MINOR ethics concerns only"]

**Final Justification:**

This is a helpful benchmark that will support other research, and the paper describes novel and effective techniques for constructing such datasets, which may be of independent interest.  I think it's a reasonable contribution to the literature.

**Limitations:**

Satisfactory.

**Quality:**

4

**Strengths And Weaknesses:**

Strengths:
- Techniques to construct training data and
  evaluation benchmarks.  (Not clear if a specific
  benchmark will be released to others?)
- Evaluation of effectiveness of existing LLMs
  and agents at two important security tasks.
- The paper is well written, very clear, easy
  to read.
- The problem is relevant.  Model providers
  will likely care.  It also sheds light on
  the capabilities of existing models and might
  stimulate new research.

Weaknesses:
- The tool reduces the cost of creating such
  a benchmark by about 2x.  Does that matter?
  Is it worth to spend months of a researcher's
  time, to save about $200?  I'm not sure.
- The analysis of ways existing agents fail
  could be a bit clearer.  It's hard to extract
  insights from the current writing.

Will you publicly release the benchmark?  Not just
the code/framework that could be used to create a
benchmark, but your benchmark of 200 issues?  I think
that would be valuable to the community.

I think it might be confusing to name your tool/framework
SEC-bench.  A name like SEC-bench would normally be used
for the benchmark, not the tool used to generate the
benchmark.  You might pick a different name for the
tool/framework/technique used to generate the benchmark.

l285: Please clarify the meaning and criteria for each of
the four categories.  For instance, what's a failed patch?
If a patch doesn't compile doesn't that mean it is incorrect?
What's the difference between incorrect vs failed?

l292: Why would integrating with git avoid incorrect
patches?  I can see why it might avoid situations where
it generates something that is not a valid patch and
thus cannot be merged, but that's different from a
patch being incorrect.

l341: I don't understand how OSS-Fuzz is relevant here
or what analogy is being drawn.

---

> ### Author Rebuttal · Authors · 2025-07-29
>
> We appreciate your insightful comments and have addressed each point in detail below.
>
> ### **Q1. The significance of reducing the cost**
> We're not sure which specific `2x` reduction you're referring to in our paper. If you could please clarify which section or metric you're asking about, we would be happy to provide a more detailed response about the significance of this cost reduction.
>
> ### **Q2. Insights to improve code agents**
> We summarize several insights and lessons that can benefit future agent design.
>
> - **Proper Context Management.** No Patch is usually caused by the large code context that quickly exceeds the context limitation of LLMs. For security tasks, the sanitizer output, compilation output, and error reports can contain lots of tokens. Even though SEC-bench provides harness for streamlining the analysis process, several jobs such as building the project and triggering the vulnerability sometimes trigger huge amounts of context. Better managing these output contexts and extracting the main information from them can reduce token consumption and help the agent focus more on the vulnerability reasoning process.
> - **Better Program Analysis Strategy.** One main reason for the low PoC generation performance is that many codebases contain numerous files, making it difficult for agents to effectively analyze data flows. Agents spend considerable resources understanding how data travels from user-controllable inputs to memory access violation points. Writing correct PoCs requires precise data flow tracking and understanding the conditions that must be satisfied. Developing data flow analysis specialized language models or agent scaffolds optimized for tracking these relationships represents a promising research direction.
> - **Proper and Task-Specific Tool Usage.** In our case study, we found that some tools are important, yet current agent scaffolds cannot properly use all available tools. For example, Aider integrates the Git tool and can utilize it to check the format of the output diff file. As a result, Aider does not have IP failure cases. In the PoC generation tasks, the debugger is very helpful for crafting precise vulnerability payloads. However, none of the existing agents can properly utilize this tool to interact with the vulnerable program. We believe that in future security agent design, guiding the agent to make full use of these task-related tools is essential, and a proper strategy will significantly benefit performance on security-related tasks.
>
> ### **Q3. Release of the benchmark**
> Yes, we will release both the benchmark and dataset along with the code and leaderboard
>
> ### **Q4. SEC-bench naming**
> Thanks for raising this concern. Our naming strategy is similar to SWE-bench, which is known as a benchmark for software engineering tasks but also includes dedicated scripts for collecting Github issues with test units. The key difference is that our framework implements a more sophisticated multi-agent scaffold (SecVerifier) for reliable software vulnerability verification, while SWE-bench uses simpler automated scripts for data collection. SEC-bench serves as both the name of our benchmark dataset and the broader framework for evaluating LLM agents on software security tasks. Within this framework, SecVerifier is a specialized multi-agent scaffold specifically designed for reproducing high-quality vulnerabilities. Our SEC-bench framework effectively combines both the benchmark dataset and the methodology for collecting and verifying it, which we believe provides clarity and consistency for users of our system.
>
> ### **Q5. Clarify the meaning and criteria for each of the four categories (L285)**
> Thank you for pointing this out. We'll clarify the failure reasons in the next version. The four categories represent distinct failure modes:
>
> 1. No Patch (NP): The agent fails to output any patch file.
> 2. Incorrect Patch (IP): The agent's output doesn't follow the diff file format, causing errors when using the `git apply` command.
> 3. Compilation Error (CE): After applying the patch to the project, the code fails to compile correctly.
> 4. Failed Patch (FP): The patch compiles successfully but doesn't fix the vulnerability—the PoC still triggers the vulnerability.
>
> The key difference between IP and FP is that IP involves an incorrectly formatted diff file, while FP has correct formatting and compiles properly but fails to address the underlying security issue.
>
> ### **Q6. Why would integrating with git avoid incorrect patches? (L292)**
> As explained, an incorrect patch (IP) refers to a patch file that doesn't follow the proper diff file format, causing errors when using the git apply command. Aider avoids this issue because it integrates the Git tool directly into its workflow. During the reasoning process, Aider attempts to use the git apply command to test its patches. If it creates an incorrectly formatted patch, it immediately encounters an error from the Git tool and tries to correct the problem. This integration explains why Aider doesn't produce incorrect patches (IP cases).
>
> ### **Q7. How is OSS-Fuzz relevant? (L341)**
> The reference to OSS-Fuzz serves two specific purposes:
>
> First, it provides a successful precedent for benchmark expansion. OSS-Fuzz began with C/C++ support in 2016 and methodically expanded to additional languages (Go in 2019, Rust in 2019, Python in 2020, Java in 2021, and JavaScript in 2023). We envision a similar expansion trajectory for SEC-bench.
>
> Second, and more importantly, OSS-Fuzz offers technical infrastructure we can leverage. Its continuous fuzzing framework provides reliable oracles for vulnerability verification which is a critical component of our framework. By integrating with OSS-Fuzz's infrastructure, we can more efficiently expand SEC-bench while maintaining verification quality.
>
> For vulnerability types beyond memory safety issues (which sanitizers effectively detect), we plan to incorporate static analysis tools like CodeQL, Joern, and Semgrep to cover logical vulnerabilities. This combination will enable SEC-bench to provide comprehensive security testing across multiple languages and vulnerability types.

---

> ### Comment · Reviewer_myBf · 2025-08-04
> **Reactions to rebuttal**
>
> Thank you for the detailed responses.  I'm encouraged to hear that the benchmark and dataset will be made publicly available.  That increases my enthusiasm for the paper.
>
> **Cost**
>
> I apologize if the $2 \times$ figure is not correct.  I probably was confused.  Perhaps you can help me understand how much reduction in cost SEC-bench achieves, compared to prior work?  The introduction says SEC-bench cost USD 0.87 per instance, representing a 85.7% improvement over CodeAct (l 76-77).  I'm not sure how to interpret that.  Perhaps the implication is that using CodeAct to generate such a benchmark would cost USD 6.08 per instance?  In which case SEC-bench reduces the cost of generating such a benchmark by $7 \times$?  Is that correct?  Is it even valid to assume that the key contribution of SEC-bench is reducing the cost of constructing such a benchmark?  Could such a benchmark have been constructed by earlier methods, or is this an entirely new capability?
>
> If that's correct, is reducing the total cost from USD 1216 to USD 174 significant?  Is it worth an entire paper's worth of effort to save about USD 1000?  Probably you spent far more than USD 1000 in labor to do this research.  Perhaps you can make the case that it is worthwhile?  e.g., because these techniques will be of independent interest, or there is a need to generate many such benchmarks, or model providers could use them to generate a large amount of training data?  Perhaps you can help walk me through the (economic) argument for why this work is important.  Or perhaps I am starting from a faulty premise and you can help me understand a better way to think about the value and contribution of SEC-bench.
>
> **Insights**
>
> Thank you for the insights and intuition.  That's helpful.
>
> Can you walk me through how you know that agents spend considerable resources understanding dataflow?  Did you conduct manual analysis of the tool calls or reasoning chains?  If so, it would be interesting to report in more detail on the results of that examination.
>
> How do you know that the agents can't properly use the debugger tool?  Can you provide any examples or case studies of how that fails?
>
> **Categories**
>
> Thanks for the detailed explanation.  That helps.  I think I was misled by the name "Incorrect Patch", which made me think that it is semantically incorrect, e.g., fails to correctly fix the vulnerability or causes functional regressions or something; and by "Failed Patch", which made me think that attempts to apply the patch failed.  As a result, I got those two backwards.  Perhaps you might consider other names, e.g., "Malformed Patch" or "Invalid Patch" or "Improper Format" instead of "Incorrect Patch", and "Incorrect Patch" or "Still Vulnerable" instead of "Failed Patch".  It's up to you, I'm not pressuring you to make any changes, I'm mentioning this in case it might help other readers develop the appropriate interpretation.
>
> **Scope and future work**
>
> I am not persuaded about the proposed directions for expanding beyond memory safety vulnerabilities and to other programming languages.  Memory safety vulnerabilities enable one to verify a claimed vulnerability and PoC with high confidence/accuracy, by using a sanitizer.  In contrast, existing open source static analysis tools (Joern, CodeQL, Semgrep) are much less accurate; they have many false positives and false negatives.
>
> I don't treat this as a positive or negative of the paper.  I think the relevance is that the current approach seems to have a fundamental limitation, that it is not easy to expand it beyond memory safety vulnerabilities to all vulnerabilities that might appear.  (It may be possible, but I expect it would require new innovations and be a research project in its own right.)  I think it would be best if the paper acknowledged this limitation clearly.  I see multiple reviews making similar remarks, so I think it would be best to own the limitation and disclose it forthrightly early in the paper, rather than suggesting it can be fixed without much difficulty in future work.

---

> ### Author Response · Authors · 2025-08-05
>
> Thanks for your follow-up questions! Our answers for each point are below:
>
> ### **Cost**
>
> There appears to be a misunderstanding about the relationship between SEC-bench and CodeAct. SEC-bench is not primarily focused on improving CodeAct or reducing costs. Rather, SEC-bench introduces a novel framework to automatically construct interactive benchmarks for security agents based on real-world CVE vulnerabilities - a capability that didn't previously exist.
>
> The primary value of SEC-bench lies in providing the first automated framework for creating high-quality security benchmarks from real-world vulnerabilities, enabling systematic evaluation of LLM security capabilities at scale, and revealing critical insights into current agent limitations in security tasks.
>
> ### **Insights**
>
> We conducted detailed analysis of the agent trajectories to understand how they handle security tasks. For PoC generation, we observed that agents spend significant time on dataflow analysis by plotting the probability distribution of tool usage across turns in the form of density plots. The `open` tool (used for reading code files) maintains consistently high usage (around 0.2 proportion) throughout the trajectory, contrasting with patterns observed in the SWE-agent [a] paper (Figure 17). This sustained file reading indicates agents are continuously analyzing the codebase to understand complex data flows required for PoC generation.
>
> Regarding debugger tool usage, we clarify that current state-of-the-art agents aren't equipped with dynamic debugging tools that would be particularly helpful for precise PoC generation. This isn't a failure of the evaluated code agents, but rather highlights a design gap in existing agent architectures for security tasks. Our finding suggests that future security agents would benefit significantly from specialized debugging tools that can provide runtime information about vulnerable programs.
>
> Based on your insightful comments, we will add more detailed trajectory analysis for both vulnerability reproduction and patching tasks to provide clearer insights for future security agent design.
>
> [a] SWE-agent: Agent-Computer Interfaces Enable Automated Software Engineering, Yang et al.
>
> ### **Categories**
>
> Thank you for this helpful suggestion. We will implement your recommended terminology changes to improve clarity. Specifically, we will rename "Incorrect Patch" to "Improper Format" and "Failed Patch" to "Still Vulnerable." These more precise terms will help readers better understand the nature of the different patch outcomes in our evaluation.
>
> ### **Scope and Future Works**
>
> We acknowledge that memory safety vulnerabilities represent only one category of software security issues, albeit a critical one. We agree with your assessment and will explicitly acknowledge this limitation early in our paper.
>
> While SEC-bench currently focuses on memory safety vulnerabilities, our approach leverages established tools like sanitizers that provide objective verification. As you correctly pointed out, expanding beyond memory safety presents significant challenges due to the lower accuracy of existing static analysis tools.
>
> However, we want to clarify our future work direction. Similar to how OSS-Fuzz has adopted variant sanitizers for Java (Jazzer) and Python (Atheris), our framework could integrate these established tools as oracles for other languages. For non-memory safety vulnerabilities, we can adapt specialized code query tools like Joern and CodeQL where our SecVerifier operates. To reduce false positives, we would instruct SecVerifier to create precise, vulnerability-specific queries rather than general CWE-type detections. Note that our SecVerifier scaffold can be easily adapted to this new task by revising prompts and equipping it with code query compilers.
>
> That being said, we appreciate your suggestion and will modify our paper to clearly state this fundamental limitation and provide a more nuanced discussion of potential expansion paths that would require significant research innovation.

---

> > ### Comment · Reviewer_myBf · 2025-08-06
> > **Thank you**
> >
> > Thank you for the thoughtful remarks.  I'm generally satisfied with the discussion.
> >
> > Cost: I suggest revising the paper to change/remove the reference to the cost comparison in the introduction, since that contributed to my misunderstanding, and I'm now not sure what point was being made there or why it is so significant that it belongs in the introduction.
> >
> > Insights: I don't think we can conclude that the agent is tracing complex dataflow, based on the observation that it uses the open tool.  I imagine there are other explanations, right?  Perhaps the agent is tracing complex control flow, or gather other context, or there are other reasons to use the open tool?  I suspect you need to look into this in a bit more detail before you can make such a statement.
> >
> > Future work: I think that is a reasonable direction to pursue in a follow-on paper, and will be valuable if it is successful.  It seems more challenging and an open question whether it will be successful.  With memory safety, we have a sanitizer that, if it triggers, provides ground truth that there is a vulnerability.  This enables you to construct a benchmark that is verified.  In contrast, if you ask a LLM to construct a CodeQL query for this specific vulnerability, I assume the query can have errors, so it seems to me like any dataset you construct won't be fully verified -- I presume you'll have to measure how accurate the resulting dataset is.  In any case, since you plan to mention the limitations in this paper, I'm satisfied.

---

### Official Review · Reviewer_Zq8U · 2025-07-03

**Clarity:** 4
**Significance:** 4
**Originality:** 4
**Rating:** 5
**Confidence:** 4

**Summary:**

This paper proposes and implements an automated benchmark framework SEC-bench for evaluating capabilities of LLM agents in real-world software security engineering tasks. Existing security benchmarks suffer from severe flaws, such as relying on artificially constructed challeges, using synthetic data, or including a large number of real-world vulnerabilities that are difficult to reproduce or have incorrect labels, which prevents accurate measurement of LLM agents' performance in actual security work.

SEC-bench introduces an innovative, multi-agent-based automated verification process (SECVERIFIER). This process decomposes complex vulnerability reproduction and verification tasks into three subtasks executed by specialized LLM agents (Builder, Exploiter, Fixer), systematically handling CVE instances collected from public databases (such as OSV). It can automatically configure isolated compilation and runtime environments (Docker), reproduce vulnerabilities (generate working Proof-of-Concepts), and verify patches. The framework uses the reports of memory safety sanitizers as an "oracle" to ensure the objectivity and reliability of the verification process.

The main contributions of this paper are: 1) proposing the first general framework capable of automatically and scalably constructing high-quality security benchmarks from real-world vulnerability reports; 2) building the SEC-bench dataset and accompanying evaluation tasks; 3) revealing through extensive experiments the current shortcomings of LLM agents in real-world security scenarios, thereby guiding the development of the field.

**Questions:**

Question 1: The reliability foundation of the entire SEC-bench framework is the memory safety sanitizer. This is a clever engineering decision, but it also introduces a profound methodological bias. The sanitizer primarily detects specific categories of vulnerabilities (such as buffer overflows, use-after-free), while it is completely powerless against other equally common and dangerous security vulnerabilities (such as logical errors, injection vulnerabilities, improper access control, cryptographic misuse, etc.). Therefore, SEC-bench is essentially a benchmark for memory safety vulnerability fixes, rather than a universal "software security" benchmark. Naming and promoting this benchmark as a general "Software Security Task" benchmark might mislead the community? Over-optimizing agents that perform well on this benchmark might give rise to a batch of "one-trick ponies" that only know how to fix "bugs reportable by sanitizers" and are helpless against other types of vulnerabilities? Would this undermine its claimed "realism"?

Suggestions: The authors should more explicitly acknowledge this limitation in the paper and provide an in-depth discussion on how this design choice shaped the distribution of vulnerability types in the dataset. A good response should analyze the distribution of vulnerability types (CWE) in SEC-bench compared to all reported vulnerability types in real-world C/C++ projects, and discuss how to expand the coverage of the benchmark in the future by introducing other static/dynamic analysis tools as new "de facto standards." This will demonstrate the authors' deep understanding of the limitations of the methodology.

---

Question 2: The Exploiter Agent is the bottleneck in the entire process (success rate of 39.4%), and also the most exciting part from a technical perspective. The authors mention that when there is no existing PoC, the intelligence will "generate one from scratch by analyzing the root cause, vulnerability patterns, and affected code paths."  In the successful verification cases of SECVERIFIER , what proportion of PoCs were completed by simply adjusting the example code in the bug report, and how many were truly "creating from scratch"? If the latter really occurred, what kind of reasoning process did the LLM demonstrate? Did it simply provide input that caused a crash, or did it construct complex payloads with specific structures (such as meeting specific offsets and lengths)?

Suggestions: The author should provide more detailed case studies or data breakdowns. For example, categorize the successful cases of Exploiter into "PoC adaptation" and "PoC generation from scratch," and provide the respective counts. For the latter, provide a specific and convincing case that demonstrates the complete thought process of the LLM, which will greatly enhance the technical depth and credibility of the paper. If the majority of successful cases belong to "adaptation," the wording should be adjusted to better reflect the actual situation.

---

Question 3: The ablation experiments in Table 3 strongly demonstrate that SECVERIFIER (multi-agent) outperforms CODEACT (single-agent). The authors attribute this to the multi-agent method. However, this advantage may stem from two coupled factors: 1) the structural advantages of task decomposition itself; 2) the prompts designed for each specialized agent, which are more context-focused. A prompt designed for a "Builder" would contain more specific instructions and context about compilation compared to a generic "Please verify this CVE" prompt. Can we decouple these two? A well-designed single agent, if given a highly structured, step-by-step "Chain-of-Thought" prompt, requiring it to sequentially play the roles of "Builder," "Exploiter," and "Fixer," and providing specific instructions for each step, can it achieve performance comparable to a multi-agent system?

Suggestions: This is a profound research question. The authors could conduct a supplementary experiment or delve deeper into this issue in the discussion. A strong response would acknowledge this ambiguity and discuss its implications for future agent design: whether we are pursuing multiple independent agents or a single, more powerful agent capable of efficient, structured role-playing within a single conversation?

**Ethical Concerns:**

["NO or VERY MINOR ethics concerns only"]

**Final Justification:**

This paper presents complete experiments and comprehensive context. The authors conducted the rebuttal very seriously and responsibly, clearly addressing the concerns I raised as well as clarifying the ambiguities in the original text. Therefore, I believe this is a very thorough and complete paper, and it should be accepted.

**Limitations:**

The mentioned limitations are honest and important! However, there might be several extras limitations:
- Selection Bias in Benchmark Construction: As I mentioned in the weakness of the "Quality" section, the author should explicitly discuss the potential selection bias introduced by the 22.3% success rate, and its impact on the representativeness of the SEC-bench dataset.

- Potential Negative Social Impact: The Exploiter Agent in SECVERIFIER is essentially an automated vulnerability exploitation tool. The authors should discuss the possibility of this technology being misused by malicious actors. For example, a powerful Exploiter Agent could be used to quickly generate PoCs for newly discovered 0-day vulnerabilities, thereby accelerating cyberattacks.

**Paper Formatting Concerns:**

All formulas, illustrations and tables are very clear with accurate labels and easy to understand. The format and writing quality are great!

**Quality:**

3

**Strengths And Weaknesses:**

Quality

Strength: 1. Breaking down complex tasks into three stages—compilation-exploitation-repair—and assigning specialized agents to handle them is a reasonable abstraction for addressing this complex problem. 2. Choosing memory-safe sanitizers as the de facto standard for vulnerability trigger and fix verification is an extremely important advantage. It avoids the subjectivity and inconsistency of manual judgment, making the entire benchmark construction and evaluation process reproducible and trustworthy.

Weaknesses: The success rate of SECVERIFIER (22.3%) is commendable in difficult tasks, but this also means that the 200 instances successfully validated by the framework were "filtered" from 898 candidate instances. This may inadvertently introduce selection bias, i.e., SEC-bench may consist mainly of vulnerabilities that are "easier to process with automated workflows." This could lead to future agents optimized for SEC-bench having their capabilities limited to solving such structured or relatively complete vulnerabilities, while lacking generalization ability for more chaotic, complex real-world problems.

---

Clarity

Strength: The organization of the paper is very clear, from the motivation of the problem, the limitations of existing work, to framework design, experimental evaluation, and even case analysis of failures, the logical chain is complete.

Weakness: None

---

Significance

Strengths: 1. This paper accurately identifies the shortcomings of existing work and provides a feasible solution. This work will push the community to shift focus from general coding capabilities to more challenging safety capabilities. 2. the evaluation results (PoC generation < 13%, vulnerability fixes < 32%) contrast sharply with general software engineering benchmarks (SWE-bench > 60%), which is a very important discovery. It sends a clear message to the entire field: the success of LLM in general coding tasks cannot be directly extrapolated to the security domain.

Weakness: None

---

Originality

Strengths: Although multi-agent systems and code generation/repair have been studied, the concept of using multi-agent systems for automated construction and verification of security benchmarks is entirely new.

Weakness: None

---

> ### Author Rebuttal · Authors · 2025-07-29
>
> Thank you for your valuable feedback. Please find our point-by-point responses below:
>
> ### **Q1. SEC-bench is essentially a benchmark for memory safety vulnerability, rather than  a universal "software security" benchmark.**
> We acknowledge the concern regarding the scope of SEC-bench being primarily focused on memory safety vulnerabilities. We will follow your suggestion to explicitly acknowledge this limitation in in the revision. Our design choice was deliberate and driven by our need for a fully automatic, scalable, and trustworthy oracle to evaluate vulnerability fixes. Memory safety sanitizers provide such reliability for an important class of vulnerabilities in C/C++ codebases.
>
> We follow your suggestion and analyze the CWE of SEC-bench. We compare the SEC-bench CWE with real-world C/C++ vulnerabilities from 2020-2024. As demonstrated in our tables, there is substantial alignment between SEC-bench's coverage and actual vulnerability distributions in the wild, with 8 out of 10 top CWE types in SEC-bench matching those found in real-world C/C++ projects. Additionally, most SEC-bench instances have high to critical CVSS scores (7.0-10.0), indicating that our benchmark captures security-critical vulnerabilities.
>
> Regarding extension to other vulnerability types and programming languages, our framework provides a foundation that can be adapted as **reliable oracles** become available. We have plans to extend support to:
>
> - Other languages via tools like Jazzer (Java) and Atheris (Python) that provide sanitizer-like capabilities
> - Logical vulnerabilities using static analysis tools such as Joern, CodeQL, and Semgrep
> - Multi-agent approaches where specialized agents write tailored code queries for specific vulnerability patterns
>
> We see parallels with SWE-bench's evolution, which began with Python GitHub issues but has expanded significantly through community efforts. SEC-bench similarly provides a strong foundation that security researchers can build upon to evaluate systems across a broader range of security vulnerabilities over time. Notably, SEC-bench provides a robust multi-agent framework for developing benchmarks for diverse software security tasks with dependable oracles. We believe this approach balances methodological rigor with practical utility while acknowledging the current limitations of our benchmark. We will revise the paper to make these points explicit and provide a roadmap for expanding vulnerability coverage in future iterations.
>
> **CWE distribution on real-world C/C++ projects**
> | CWE | Count | Percentage|
> | ---- | ---- | ----|
> CWE-787	| 1054	| 16.3%
> CWE-125	| 649	| 10.1%
> CWE-476	| 479	| 7.4%
> CWE-120	| 308	| 4.8%
> CWE-190	| 294	| 4.6%
> CWE-416	| 253	| 3.9%
> CWE-20	| 207	| 3.2%
> CWE-617	| 194	| 3.0%
> CWE-122	| 176	| 2.7%
> CWE-119	| 173	| 2.7%
>
> **CWE distribution on SEC-bench**
> | CWE | Count | Percentage|
> | ---- | ---- | ----|
> CWE-125	| 45 | 	23.81%
> CWE-787	| 44 | 	23.28%
> CWE-476	| 33 | 	17.46%
> CWE-416	| 21 | 	11.11%
> CWE-119	| 8	 | 4.23%
> CWE-772	| 7	 | 3.70%
> CWE-401	| 7	 | 3.70%
> CWE-122	| 7	 | 3.70%
> CWE-190	| 5	 | 2.65%
> CWE-120	| 3	 | 1.59%
>
> ### **Q2. The proportion of instances whose PoC are generated from scratch in the exploit agent.**
> Thank you for your valuable feedback on our SecVerifier system, particularly regarding the Exploiter Agent's capability to generate PoCs from scratch.
>
> We follow your suggestion to analyze the trajectory of Exploiter. We find that, out of 289 instances where a PoC was successfully crafted, only 3 were generated completely from scratch using the GPT-4o model. This yields a ratio of approximately 99:1 for "PoC adaptation" versus "PoC generation from scratch." This low rate of from-scratch generation was significantly impacted by our computational constraints—we limited each instance to a maximum of 75 iterations due to cost considerations, which restricted our ExploiterAgent's ability to fully understand root causes and craft valid PoC inputs.
>
> Following your suggestions, we will modify the wording "When no available PoC is found, the agent generates one from scratch" in our paper to better reflect the actual situation while pointing toward promising future research directions.
>
> We also show one successful case where the PoC was successfully crafted in our rebuttal. Generally, our ExploitAgent follows these steps:
>
> 1. Read vulnerable code snippets across files to understand root causes
> 2. Use a Python tool to craft precise input bytes to trigger the vulnerability
> 3. Use our harness to trigger the vulnerability
> 4. Check the results and complete the task if it successfully triggers the same sanitizer error; if not, return to step 1 and continue analysis
>
> To illustrate the reasoning process demonstrated by our system, we present the example of the `libplist.cve-2017-7982` instance, a heap buffer overflow vulnerability in `/src/libplist/src/bplist.c:733`. The process begins by reading functions from the call stack in the sanitizer logs from the bug report. The agent reads relevant files like `/src/libplist/src/bplist.c` and `/src/libplist/tools/plistutil.c` to understand the root causes. Then, it creates the first PoC input using a Jupyter Python tool for precise byte offset calculation.
>
> After failing in the first attempt, the agent demonstrates its reasoning capability:
>
> *"This could be due to the PoC not being crafted correctly to trigger the vulnerability. I need to ensure that the PoC file is structured in a way that causes the heap-buffer-overflow as described in the bug report."*
>
> Following several iterative failures, it rereads the vulnerable file and engages in deeper analysis:
>
> *"The code at line 733 in* **`bplist.c`** *is where the heap-buffer-overflow occurs. The condition* **`if (index1 &lt; num_objects)`** *is meant to ensure that the index is within bounds, but the calculation of* **`index1`** *might be incorrect due to integer overflow or incorrect data in* **`data->buff`***. The PoC needs to simulate a scenario where* **`index1`** *is calculated to be out of bounds, leading to the overflow. I will adjust the PoC to ensure that* **`data->buff`** *is crafted in a way that causes* **`index1`** *to exceed* **`num_objects`***. This might involve setting specific values in the buffer to manipulate the calculation of* **`index1`***. I will update the PoC accordingly."*
>
> This reasoning ultimately leads to a successfully crafted complete PoC input.
>
> Based on these findings, we will add a dedicated section to the appendix that thoroughly documents the successful cases where PoC inputs were generated from scratch. This will provide valuable insights into how to effectively elicit advanced reasoning capabilities from LLMs in security vulnerability analysis. We believe that with increased iteration limits and more sophisticated reasoning frameworks, the proportion of from-scratch PoC generation could be significantly improved in future research.
>
> ### **Q3. Deeper analysis on why SecVerifier (multi-agent) outperforms CodeACT (single-agent)**
> We appreciate the reviewer's insightful question regarding the performance advantage of our multi-agent system (SecVerifier) over the single-agent approach (CodeACT).  This is an important point about decoupling two potential factors: the structural benefits of task decomposition versus the specialized prompts designed for each agent. Actually, in our ablation study, we have carefully controlled for this exact concern. The single-agent system was provided with the same comprehensive prompt that contained all instructions given to the specialized agents, with only structural differences. Specifically, the single-agent prompt included the same metadata and instructions for all three phases (build, exploit, fix), while the multi-agent system had these instructions distributed across specialized agents.
>
> The key difference between multi-agent and single-agent in our ablation study was that single agents sometimes terminated tasks prematurely with incorrect results, primarily due to failures in context management when handling complex vulnerability verification tasks. Our multi-agent approach solved this by dividing complex tasks among specialized agents and implementing a manager agent to validate results and provide feedback for iteration. Both approaches were given the same maximum number of iterations (75) to ensure fair comparison.
>
> In the revised version, we will add a brief explanation in the main text to elaborate on these details. We will include comprehensive details in the appendix, including the prompts used for both approaches in our ablation study.
>
> This finding has important implications for future agent design, suggesting that while well-crafted prompts are essential, the architecture of multiple specialized agents working in concert offers substantive advantages for complex tasks requiring sustained attention across multiple phases.
>
> ### **Q4. Extra limitations: selection bias and potential negative social impact**
> We appreciate you highlighting these additional limitations. We agree with your feedback and will incorporate these points into the next version of the paper.

---

### Official Review · Reviewer_s6x1 · 2025-07-05

**Clarity:** 3
**Significance:** 2
**Originality:** 2
**Rating:** 4
**Confidence:** 3

**Summary:**

This paper proposes SEC-bench, a benchmarking framework designed to evaluate LM agents on real-world software security engineering tasks. The method addresses the limitations of existing security benchmarks which rely on synthetic tasks or simplified vulnerability cases. SEC-bench uses an effective multi-agent scaffold that automatically constructs code repositories, reproduces vulnerabilities in isolated environments, and generates gold patches for evaluation at just $0.87 per instance. The framework is tested with actual CVE instances from open-source projects with two tasks: Proof-of-Concept (PoC) generation (12.5% success rate), Vulnerability patching (31.5% success rate).

Overall, SEC-bench represents a valuable contribution to security engineering benchmarking of LM agents. The low success rates on PoC generation and patching highlight the significant challenges in this domain. Although the limitations in current language support is a major obstacle to larger impact, the framework's cost-effective automation make it a promising foundation for future work.

**Questions:**

- Can you elaborate on how the framework can be extended to support more languages and more vulnerability types?

**Ethical Concerns:**

["NO or VERY MINOR ethics concerns only"]

**Final Justification:**

I stand by my recommendation to accept the paper with minor revisions. The paper presents a solid benchmarks with some specific limitations. In the rebuttal, the authors present clear plans to fix those limitations.

**Limitations:**

Yes

**Quality:**

3

**Strengths And Weaknesses:**

**Strengths**
- The paper tackles a real gap in security benchmarking for LMs by moving from synthetic vulnerabilities to real-world problems.
- The agent scaffolding of the framework can be a good foundation for future work. The architecture achieves 85.7% improvement over single-agent baseline, and remains cost-effective, costing $0.87 per verified instance.
- The paper uses the proposed framework to evaluate multiple state-of-the-art LM agents and revealed significant performance gaps.

**Weaknesses**
- Limited Language Support. The framework only supports C/C++ and is limited to memory safety vulnerabilities detectable by sanitizers, excluding many modern languages and web vulnerabilities.
- The current low success rate (22.3% verification, 12.5% for PoC generation, 31.5% for patching) raises concerns about the framework's validity for benchmarking models. While the authors mention other possible tasks like fuzzing and vulnerability discovery, only two tasks are actually implemented and evaluated.
- The benchmark currently only contains 200 verified instances from 898 candidates, despite the cost-effective automation.

---

> ### Author Rebuttal · Authors · 2025-07-29
>
> Thank you for your insightful comments. We have addressed each of your points in the following responses:
>
> ### **Q1. Limited language support**
> For the support of more languages, we can incorporate other languages' sanitizer-like tools to include the vulnerabilities of these languages into SEC-bench. As illustrated in L102, SEC-bench utilizes sanitizers as the vulnerability oracle to reproduce C/C++ vulnerabilities. Other languages also have these kinds of tools, such as Jazzer (Java) and Atheris (Python). Incorporating these tools as oracles can extend SEC-bench's coverage to other languages.
>
> For the support of more vulnerability types, SEC-bench can include logical vulnerabilities using static analysis tools such as Joern, CodeQL, and Semgrep.
>
> ### **Q2. Low success rate**
> Thank you for raising this important concern. We understand how the low success rates might raise questions about the benchmark's validity. We believe the relatively low success rates actually reveal the genuine difficulty of these security tasks for current LLM agents, rather than indicating flaws in the benchmark itself.
>
> The benchmark's validity stems from our use of memory-safety sanitizers as reliable, deterministic oracles to verify the existence of vulnerabilities in codebases. These sanitizers provide ground truth independent of any agent's performance. In other words, each instance in our benchmark represents a real, verified vulnerability with a corresponding sanitizer-detectable issue - regardless of whether current agents can successfully address it.
>
> This challenging nature of our benchmark actually makes it more valuable for driving progress in AI security capabilities. The gap between current performance and perfect scores highlights areas where improvement is needed, and provides a clear metric for measuring future advances.
>
> As for the extensibility to other tasks, we can readily extend SEC-bench to additional critical software security challenges like fuzz driver generation and vulnerability discovery using the same sanitizer-based verification approach. The core strength of SEC-bench is that we can build reliable datasets for diverse security tasks when we have dependable oracles that can deterministically and reliably identify security vulnerabilities.
>
> ### **Q3. Only contains 200 verified instances**
> As we highlight in our paper (L54-L62), the main challenge in creating reproducible security benchmarks is acquiring reliable Proof-of-Concept (PoC) artifacts. This difficulty is well-documented in security research. According to Mu et al. [a], vulnerability reports from popular security forums have extremely low reproduction rates (4.5%-43.8%) due to missing information, ambiguous descriptions, and environment dependencies. In this context, our ExploiterAgent's success rate of 39.4% is quite strong, falling at the upper end of what prior research has found possible even with manual efforts.
>
> Regarding benchmark size, we prioritize quality over quantity for several reasons:
>
> 1. Each of our 200 instances is thoroughly verified, ensuring that researchers can trust the ground truth of each vulnerability
> 2. Security benchmark evaluation with agent-based approaches is computationally intensive and expensive
> 3. Our carefully curated dataset covers diverse projects and vulnerability types that represent real-world scenarios
>
> We have concrete plans to expand SEC-bench by: 1) integrating OSS-Fuzz bug instances which already have verification infrastructure in place, and 2) extending to other vulnerability types and programming languages while maintaining our rigorous validation standards.
>
> [a] Understanding the Reproducibility of Crowd-reported Security Vulnerabilities (USENIX Security 2018), Dongliang Mu et al.

---

> > ### Comment · Reviewer_s6x1 · 2025-08-05
> > **Maintaining recommendation to accept**
> >
> > Thanks for the response. Seems clear to me that the authors have good ideas to improve the benchmark and fix its weaknesses. I maintain my recommendation to accept the paper.

---

### Official Review · Reviewer_RJFD · 2025-07-07

**Clarity:** 3
**Significance:** 3
**Originality:** 3
**Rating:** 4
**Confidence:** 3

**Summary:**

This paper introduces SEC-bench, a novel and fully automated framework for benchmarking the performance of Large Language Model (LLM) agents on real-world software security tasks. The authors argue that existing security benchmarks are inadequate, as they often rely on synthetic challenges, simplified vulnerability datasets, or suffer from poor reproducibility, failing to capture the complexity and ambiguity that security engineers face in practice.

**Questions:**

See weaknesses above.

**Ethical Concerns:**

["NO or VERY MINOR ethics concerns only"]

**Final Justification:**

Based on the reviewer's clarification confirming that C/C++ memory safety vulnerabilities are the core focus of the current work and SEC-bench's alignment with the AIxCC competition scope, I will maintain my positive score on the paper. I look forward to seeing the proposed future expansions (more languages/vulnerability types) in subsequent work.

**Limitations:**

Yes

**Quality:**

3

**Strengths And Weaknesses:**

Strengths:
1. he work addresses a critical and timely problem: the rigorous evaluation of LLM agents' security capabilities. The proposed SEC-bench framework is highly novel, presented as the first end-to-end automated system for constructing benchmarks from authentic, real-world vulnerabilities.
2. he design of SEC-bench is robust. The multi-agent architecture of SECVERIFIER, which decomposes the complex verification process into specialized sub-tasks (building, exploiting, fixing), is a logical and effective approach.

Weaknesses:
1. The framework's current scope is primarily limited to memory safety vulnerabilities within C/C++ projects. This is far from sufficient for generalization to real applications.

---

> ### Author Rebuttal · Authors · 2025-07-29
>
> We appreciate the opportunity to clarify this aspect of our work.
>
> SEC-bench currently focuses on memory safety vulnerabilities in C/C++ projects, aligning with the DARPA AIxCC challenge that evaluates LLMs' ability to detect and fix sanitizer-detected vulnerabilities. It's worth noting that the competition only considers specific types of vulnerabilities in C/C++ and Java that can be detected by sanitizers [a]. That being said, as future work, we plan to expand SEC-bench's scope to include more diverse vulnerability types and programming languages. For additional languages, we can incorporate language-specific sanitizers like Jazzer (Java) and Atheris (Python). To address more vulnerability types, SEC-bench can integrate logical vulnerabilities using static analysis tools such as Joern, CodeQL, and Semgrep.
>
> [a] Artificial Intelligence Cyber Challenge (AIxCC): AIxCC Semifinal Competition (ASC) Procedures and Scoring Guide

---

> > ### Comment · Reviewer_RJFD · 2025-08-06
> >
> > Based on the reviewer's clarification confirming that C/C++ memory safety vulnerabilities are the core focus of the current work and SEC-bench's alignment with the AIxCC competition scope, I will maintain my positive score on the paper. I look forward to seeing the proposed future expansions (more languages/vulnerability types) in subsequent work.

---

### Note · Authors · 2025-08-12

We thank all reviewers for their constructive feedback that has improved our paper. After addressing each concern, here's our consolidated response:

1. **Vulnerability Coverage**: We acknowledge that SEC-bench currently focuses primarily on memory safety vulnerabilities detectable by sanitizers. This is an intentional design choice that provides objective verification rather than a limitation in methodology. We have clarified this scope throughout the paper and discussed potential extensions to other vulnerability types using specialized verification tools.
2. **Methodology and Selection Bias**: We have addressed concerns about potential selection bias in our benchmark construction process. While our 22.3% success rate in SecVerifier might seem to favor "easier" vulnerabilities, our manual analysis confirms that the selected instances represent diverse complexity levels and vulnerability patterns. Nevertheless, we’ll add a thorough discussion of this limitation.
3. **PoC Generation Analysis**: As requested, we've conducted a detailed analysis of "from scratch" PoC generation versus adaptation of existing examples. We'll include case studies demonstrating the agent's reasoning process and the distribution of different exploitation approaches, providing greater transparency about agent capabilities.
4. **Multi-Agent vs. Single-Agent Architecture**: We'll clarify the advantages of our multi-agent approach beyond mere task decomposition. While structured prompting offers benefits, our experiments demonstrate that specialized agents with focused contexts significantly outperform even well-structured single agents. This performance gap stems from reduced context pollution and task-specific optimization. We'll also include prompt templates for both agent architectures for clarification.
5. **Agent Behavior Analysis**: We’ll elaborate on our analysis of agent trajectories to provide clearer insights into how agents approach security tasks, with particular attention to tool usage patterns during PoC generation and vulnerability patching.

Our research demonstrates the significant gap between general code capabilities and security-specific tasks. SEC-bench provides a foundation for measuring and improving agent performance on real-world security challenges. We believe this work will drive significant advancements in security-specialized agent development, creating more robust AI systems capable of addressing real-world software security challenges.

---

### Decision · Program_Chairs · 2025-09-17

**Decision:**

Accept (poster)

**Comment:**

SEC-bench introduces an automated benchmarking framework for evaluating LLM agents on real-world software security tasks. It employs a multi-agent scaffold to construct code repositories, reproduce vulnerabilities, and generate patches, with a focus on memory safety vulnerabilities in C/C++ projects. Two critical software security tasks are defined to rigorously evaluate LLM agents' capabilities: proof-of-concept (PoC) generation and vulnerability patching. Existing LLMs show low success rates in these two tasks.  The framework provides a cost-effective and robust methodology for measuring and improving agent performance in software security. The discussions with reviewers are quite comprehensive, including the current limitations in vulnerability coverage and the potential for selection bias in benchmark construction, etc.